# Gradual Weisfeiler-Leman: Slow and Steady Wins the Race

**Franka Bause**[1,2]        **Nils M. Kriege**[1,3]

[1]Faculty of Computer Science, University of Vienna, Vienna, Austria
[2]UniVie Doctoral School Computer Science, University of Vienna, Vienna, Austria
[3]Research Network Data Science, University of Vienna, Vienna, Austria
`{franka.bause, nils.kriege}@univie.ac.at`

## Abstract

The classical Weisfeiler-Leman algorithm aka color refinement is fundamental for graph learning with kernels and neural networks. Originally developed for graph isomorphism testing, the algorithm iteratively refines vertex colors. On many datasets, the stable coloring is reached after a few iterations and the optimal number of iterations for machine learning tasks is typically even lower. This suggests that the colors diverge too fast, defining a similarity that is too coarse. We generalize the concept of color refinement and propose a framework for gradual neighborhood refinement, which allows a slower convergence to the stable coloring and thus provides a more fine-grained refinement hierarchy and vertex similarity. We assign new colors by clustering vertex neighborhoods, replacing the original injective color assignment function. Our approach is used to derive new variants of existing graph kernels and to approximate the graph edit distance via optimal assignments regarding vertex similarity. We show that in both tasks, our method outperforms the original color refinement with only a moderate increase in running time advancing the state of the art.

## 1 Introduction

The (1-dimensional) Weisfeiler-Leman algorithm, also referred to as *color refinement*, iteratively refines vertex colors by encoding colors of neighbors and was originally developed as a heuristic for the graph isomorphism problem. Although it cannot distinguish some non-isomorphic graph pairs, for example strongly regular graphs, it succeeds in many cases. It is widely used as a sub-routine in isomorphism algorithms today to reduce ambiguities that have to be resolved by backtracking search [1]. It has also gained high popularity in graph learning, where the technique is used to define graph kernels [2–5] and to formalize the expressivity of graph neural networks, see the recent surveys [6, 7]. Graph kernels based on Weisfeiler-Leman refinement provide remarkable predictive performance while being computationally highly efficient. The original Weisfeiler-Leman subtree kernel [2] and its variants and extensions, e.g., [3–5], provide state-of-the-art classification accuracy on many datasets and are widely used baselines. The update scheme of the Weisfeiler-Leman algorithm is similar to the idea of neighborhood aggregation in graph neural networks (GNNs). It has been shown that (i) the expressive power of GNNs is limited by the Weisfeiler-Leman algorithm, and (ii) that GNN architectures exist that reach this expressive power [8, 9].

As a consequence of its original application, the Weisfeiler-Leman algorithm assigns discrete colors and does not distinguish minor and major differences in vertex neighborhoods. Most Weisfeiler-Leman graph kernels match vertex colors of the first few refinement steps by equality, which can be considered too rigid, since these colors encode complex neighborhood structures. In machine learning tasks, a more fine-grained differentiation appears promising. Real-world data is often noisy leading to small differences in vertex degree. Such differences get picked up by the refinement strategy of the Weisfeiler-Leman algorithm and cannot be distinguished from significant differences.

We address this problem by providing a different approach to the refinement step of the Weisfeiler-Leman algorithm: We replace the injective relabeling function with a non-injective one to gain a more

F. Bause et al., Gradual Weisfeiler-Leman: Slow and Steady Wins the Race. *Proceedings of the First Learning on Graphs Conference (LoG 2022)*, PMLR 198, Virtual Event, December 9–12, 2022.

gradual refinement of colors. This allows obtaining a finer vertex similarity measure, distinguishing between large and small changes in vertex neighborhoods with increasing radius. We characterize the set of functions that, while not necessarily injective, guarantee that the stable coloring of the original Weisfeiler-Leman algorithm is reached after a possibly higher number of iterations. Thus, our approach preserves the expressive power of the Weisfeiler-Leman algorithm. We discuss a realization of such a function and use $k$-means clustering in our experimental evaluation as an exemplary one.

**Our Contribution.**

1. We propose refining, neighborhood preserving (*renep*) functions, which generalize the concept of color refinement. This family of functions leads to the coarsest stable coloring while only incorporating direct neighborhoods.

2. We show the connections of our approach to the original Weisfeiler-Leman algorithm, as well as other vertex refinement strategies.

3. We propose two new graph kernels based on renep functions, which outperform state-of-the-art kernels on synthetic and real-world datasets, with only a moderate increase in running time.

4. We apply our new approach for approximating the graph edit distance via bipartite graph matching and show that it outperforms state-of-the-art heuristics.

## 2   Related Work

Various graph kernels based on Weisfeiler-Leman refinement have been proposed [2–5, 10]. Recent comprehensive experimental evaluations confirm their high classification accuracy on many real-world datasets [11, 12]. In both studies, the classical Weisfeiler-Leman subtree kernel [2] is shown to provide a competitive baseline. Variants based on optimal assignments [5, 10] improve the classification accuracy on some datasets and achieve excellent results overall.

Most Weisfeiler-Leman based approaches implicitly match colors by equality, which can be considered too rigid, since colors encode unfolding trees representing complex neighborhood structures. Some recent works address this problem: Yanardag and Vishwanathan [13] introduced similarities between colors using techniques inspired by natural language processing, which were subsequently refined by Narayanan et al. [14]. Schulz et al. [15] define a distance function between colors by comparing the associated unfolding trees using a tree edit distance. Based on this distance, the colors are clustered to obtain a new graph kernel. Although the tree edit distance is polynomial-time computable, the running time of the algorithm is very high. A kernel based on the Wasserstein distance of sets of unfolding trees was proposed by Fang et al. [16]. The vertices of the graphs are embedded into $\ell_1$ space using an approximation of the tree edit distance between their unfolding trees. A graph can then be seen as a distribution over those embeddings. While the function proposed is not guaranteed to be positive semidefinite, the method showed results similar to and, in some cases, exceeding state-of-the-art techniques. The running time, however, is still very high and the method is only feasible for unfolding trees of small height. These approaches define similarities between Weisfeiler-Leman colors and the associated unfolding trees. Our approach, in contrast, alters the Weisfeiler-Leman refinement procedure itself and does not rely on the computationally expensive matching of unfolding trees.

Some graph kernels use a neighborhood aggregation scheme similar in spirit to the Weisfeiler-Leman algorithm with a non-injective relabeling function. The neighborhood hash kernel [17] represents labels by bit-vectors and uses logical operations and hashing to encode neighborhoods efficiently. Shervashidze [18, Section 3.7.3.2] proposed to compress neighborhood label histograms using locality sensitive hashing allowing collisions of similar neighborhoods. Propagation kernels [19] provide a generic framework to obtain graph kernels from (neighborhood) propagation schemes by comparing label distributions after every propagation step. In contrast to these methods, our approach uses a data-dependent non-injective relabel function and guarantees that the expressive power of the Weisfeiler-Leman algorithm is reached.

Several authors proposed techniques supporting graphs with continuous attributes directly based on or inspired by the Weisfeiler-Leman algorithm. Propagation kernels were adapted to this setting by using a hash function to map continuous attributes to discrete labels [19]. The hash graph kernel framework [20] repeatedly performs such a discretization using random hash functions, applies a kernel for graphs with discrete labels to each resulting graph and combines their feature vectors. It

obtains high classification accuracies when combined with the Weisfeiler-Leman subtree kernel. The Wasserstein Weisfeiler-Leman graph kernel [3] establishes node matchings similar to the Weisfeiler-Leman optimal assignment kernel [5, 7]. The authors support continuous attributes by replacing discrete colors with real-valued vectors without guaranteeing that the resulting function is positive semidefinite. Finally, graph neural networks can be viewed as a neural version of the Weisfeiler-Leman algorithm, where continuous feature vectors replace colors, and neural networks are used to aggregate over local node neighborhoods [7–9]. None of these methods explicitly aims to slow down the Weisfeiler-Leman refinement process.

## 3   Preliminaries

In this section we provide the definitions necessary to understand our new vertex refinement algorithm. We first give a short introduction to graphs and the original Weisfeiler-Leman algorithm, before we cover graph kernels.

**Graph Theory.**   A *graph* $G = (V, E, \mu, \nu)$ consists of a set of vertices $V$, denoted by $V(G)$, a set of edges $E(G) = E \subseteq \binom{V}{2}$ between the vertices, a labeling function for the vertices $\mu \colon V \to L$, and a labeling function for the edges $\nu \colon E \to L$. The set $L$ contains categorical labels, which can be represented as natural numbers. We discuss only undirected graphs and denote an edge between $u$ and $v$ by $uv$. The set of neighbors of a vertex $v \in V$ is denoted by $N(v) = \{u \mid uv \in E\}$. A *(rooted) tree* $T$ is a simple (no self-loops or multi-edges), connected graph without cycles and with a designated node $r$ called *root*. A tree $T'$ is a *subtree* of a tree $T$, denoted by $T' \subseteq T$, iff $V(T') \subseteq V(T)$. The root of $T'$ is the node closest to the root in $T$.

A vertex *coloring* $c \colon V(G) \to \mathbb{N}_0$ of a graph $G$ is a function assigning each vertex a color. For vertices with $c(u) = c(v)$ we also write $u \approx_c v$. A coloring $\pi$ on a set $S$ is a *refinement* of (or *refines*) a coloring $\pi'$, iff $s_1 \approx_\pi s_2 \Rightarrow s_1 \approx_{\pi'} s_2$ for all $s_1$, $s_2$ in $S$. We denote this by $\pi \preccurlyeq \pi'$ and write $\pi \equiv \pi'$ if $\pi \preccurlyeq \pi'$ and $\pi' \preccurlyeq \pi$. If $\pi \preccurlyeq \pi'$ and $\pi \not\equiv \pi'$, we say that $\pi$ is a *strict refinement* of $\pi'$, written $\pi \prec \pi'$. The refinement relation defines a partial ordering on the colorings.

**Color Hierarchy.**   We consider a sequence of vertex colorings $(\pi_0, \pi_1, \ldots, \pi_h)$ with $\pi_h \preccurlyeq \cdots \preccurlyeq \pi_0$ and assume that for $i \neq j$ the colors assigned by $\pi_i$ and $\pi_j$ are distinct. We can interpret such a sequence of colorings as a *color hierarchy*, i.e., a tree $\mathcal{T}_h$ that contains a node for each color $c \in \{\pi_i(v) \mid i \in \{0, \ldots, h\} \wedge v \in V(G)\}$ and an edge $(c, d)$ iff $\exists v \in V(G) \colon \pi_i(v) = c \wedge \pi_{i+1}(v) = d$. We associate each tree node with the set of vertices of $G$ having that color.[1] Here, we assume that the initial coloring is uniform. If this is not the case, we add an artificial root node and connect it to the initial colors. Likewise we insert the coloring $\pi_0 = \{V(G)\}$ as first element in the sequence of vertex colorings. An example color hierarchy is given in Figure 1.

Using this color hierarchy we can derive multiple colorings on the vertices: Choosing exactly one color on every path from the leaves to the root (or only the root), always leads to a valid coloring. The finest coloring is induced by the colors representing the leaves of the tree. Given a color hierarchy $T$, we denote this coloring (which is equal to $\pi_h$) by $\pi_T$.

**Weisfeiler-Leman Color Refinement.**   The 1-dimensional Weisfeiler-Leman (WL) algorithm or color refinement [21, 22] starts with a coloring $c_0$, where all vertices have a color representing their label (or a uniform coloring in case of unlabeled vertices). In iteration $i$, the coloring $c_i$ is obtained by assigning each vertex $v$ in $V(G)$ a new color according to the colors of its neighbors, i.e.,

$$c_{i+1}(v) = z\left(c_i(v), \{\!\!\{c_i(u) \mid u \in N(v)\}\!\!\}\right),$$

where $z \colon \mathbb{N}_0 \times \mathbb{N}_0^{\mathbb{N}_0} \to \mathbb{N}_0$ is an injective function. Figure 1 depicts the first iterations of the algorithm for an example graph.

After enough iterations the number of different colors will no longer change and this resulting coloring is called the *coarsest stable coloring*. The coarsest stable coloring is unique and always reached after at most $|V(G)| - 1$ iterations. This trivial upper bound on the number of iterations is tight [23]. In practice, however, Weisfeiler-Leman refinement converges much faster (see Appendix C).

---

[1] Here, we consider a color hierarchy for a single graph. However, given corresponding colorings on a set of graphs, a color hierarchy can be generated in the same way. In an inductive setting, where no fixed set of graphs is given, the color hierarchy contains all colors, that can be produced by the corresponding coloring algorithm.

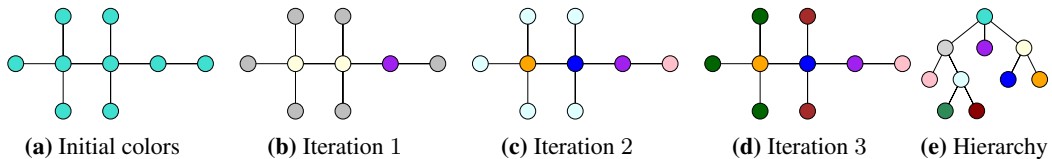

**(a)** Initial colors     **(b)** Iteration 1     **(c)** Iteration 2     **(d)** Iteration 3     **(e)** Hierarchy

**Figure 1:** Initial coloring and results of the first three iterations of the Weisfeiler-Leman algorithm. To use less colors for this example, vertices with a unique color do not get a new color. The color hierarchy shows the development of the colors over the refinement iterations.

**Graph Kernels and the Weisfeiler-Leman Subtree Kernel.**   A *kernel* on $X$ is a function $k\colon X \times X \to \mathbb{R}$, so that there exist a Hilbert space $\mathcal{H}$ and a mapping $\phi\colon X \to \mathcal{H}$ with $k(x,y) = \langle \phi(x), \phi(y) \rangle$ for all $x$, $y$ in $X$, where $\langle \cdot, \cdot \rangle$ is the inner product of $\mathcal{H}$. A *graph kernel* is a kernel on graphs, i.e., $X$ is the set of all graphs.

The Weisfeiler-Leman subtree kernel [2] with height $h$ is defined as

$$k_{ST}^h(G_1, G_2) = \sum_{i=0}^{h} \sum_{u \in V(G_1)} \sum_{v \in V(G_2)} \delta(c_i(u), c_i(v)), \tag{1}$$

where $\delta$ is the Dirac kernel (1, iff $c_i(u)$ and $c_i(v)$ are equal, and 0 otherwise). It counts the number of vertices with common colors in the two graphs up to the given bound on the number of Weisfeiler-Leman iterations.

## 4   Gradual Weisfeiler-Leman Refinement

As a different approach to the refinement step of the Weisfeiler-Leman algorithm, we essentially replace the injective relabeling function with a non-injective one. We do this by allowing vertices with differing neighbor color multisets to be assigned the same color under some conditions. Through this, the number of colors per iteration can be limited, allowing to obtain a more gradual refinement of colors. To reach the same stable coloring as the original Weisfeiler-Leman algorithm, the function has to assure that vertices with differing colors in one iteration will get differing colors in future iterations and that in each iteration at least one color is split up, if possible.

We first define the property necessary to reach the stable coloring of the original Weisfeiler-Leman algorithm and discuss connections to the original as well as other vertex refinement algorithms. Then we provide a realization of such a function by means of clustering, which is used in our experimental evaluation. Figure 2 illustrates our idea. It depicts the initial coloring, the result of the first iteration of WL and a possible result of the first iteration of the gradual Weisfeiler-Leman refinement (GWL), when restricting the maximum number of new colors to two by clustering the neighbor color multisets.

**Update Functions.**   Using the same approach as the Weisfeiler-Leman algorithm, the color of a vertex is updated iteratively according to the colors of its neighbors. Let $\mathcal{T}_i$ denote a color hierarchy belonging to $G$ and $n_i(v) = \{\!\!\{\pi_{\mathcal{T}_i}(x) \mid x \in N(v)\}\!\!\}$ the neighbor color multiset of $v$ in iteration $i$. We use a similar update strategy, but generalize it using a special type of function:

$$\forall v \in V(G)\colon c_{i+1}(v) = \pi_{\mathcal{T}_{i+1}}(v), \text{with } \mathcal{T}_{i+1} = f(G, \mathcal{T}_i),$$

where $f$ is a refining, neighborhood preserving function.

A *refining, neighborhood preserving (renep)* function $f$ maps a pair $(G, \mathcal{T}_i)$ to a tree $\mathcal{T}_{i+1}$, such that
**Condition 1.** $\mathcal{T}_i \subseteq \mathcal{T}_{i+1}$
**Condition 2.** $\mathcal{T}_i = \mathcal{T}_{i+1}$, iff $\forall v, w \in V(G) : v \approx_{\pi_{\mathcal{T}_i}} w \Rightarrow n_i(v) = n_i(w)$
**Condition 3.** $\mathcal{T}_i \subsetneq \mathcal{T}_{i+1} \Rightarrow \pi_{\mathcal{T}_{i+1}} \prec \pi_{\mathcal{T}_i}$
**Condition 4.** $\forall v, w \in V(G): (v \approx_{\pi_{\mathcal{T}_i}} w \wedge n_i(v) = n_i(w)) \Rightarrow v \approx_{\pi_{\mathcal{T}_{i+1}}} w$

The conditions assure, that the coloring $\pi_{\mathcal{T}_{i+1}}$ is a strict refinement of $\pi_{\mathcal{T}_i}$, if there exists a strict refinement: Condition 1 assures that the new coloring is a refinement of the old one. Condition 2 assures that the tree (and in turn the coloring) only stays the same, iff the stable coloring is reached,

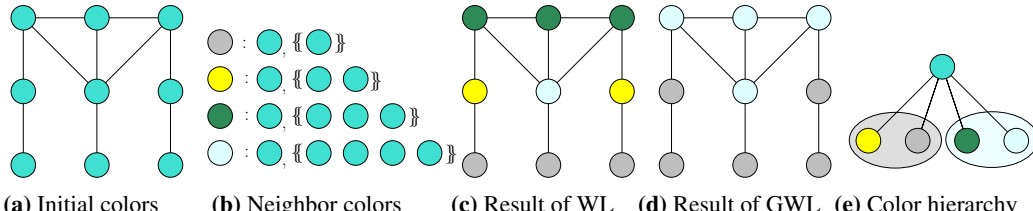

(a) Initial colors  (b) Neighbor colors  (c) Result of WL  (d) Result of GWL  (e) Color hierarchy

**Figure 2:** Initial coloring and results of the first iteration using WL and GWL refinement. We assume that the update function of GWL is a clustering algorithm, producing two clusters per old color. Vertices colored gray and yellow by WL are put into the same cluster, as well as green and light blue ones, as their neighbor color multisets only differ by one element each.

while Condition 3 assures that, if the trees are not equal, $\pi_{\mathcal{T}_{i+1}}$ is a strict refinement of $\pi_{\mathcal{T}_i}$. Without this condition it would be possible to obtain a tree, that fulfills Condition 1 but does not strictly refine the coloring (for example by adding one child to each leaf). Condition 4 assures that vertices, that are indistinguishable regarding their color and their neighbor color multiset, get the same color (as in the original Weisfeiler-Leman algorithm).

We call this new approach *gradual Weisfeiler-Leman refinement* (GWL refinement). Since $f$ is a renep function, it is assured that at least one color is split into at least two new colors, if the stable coloring is not yet reached. This property and its implications are explored in the following section.

Usually, the refinement is computed simultaneously for multiple graphs. This can be realized by using the disjoint union of all graphs as input. Note that this will have an influence on the function $f$, since refinements might differ based on the vertices involved. This is a typical case of transductive learning, because the algorithm has to run on all graphs and if a new graph is encountered, the algorithm has to run again on the enlarged graph. To counteract this, one could only run the algorithm on the training set and keep track of the coloring choices. Then for new graphs, the vertices are colored with the most similar colors.

### 4.1 Equivalence of the Stable Colorings

The gradual color refinement will never assign two vertices the same color, if their colors differed in the previous iteration, since we require the coloring to be a refinement of the previous one. We can show that the stable coloring obtained by GWL refinement using any renep function is equal to the unique coarsest stable coloring, which is obtained by the original Weisfeiler-Leman algorithm.

**Theorem 5** ([24], Proposition 3). *For every coloring $\pi$ of $V(G)$, there is a unique coarsest stable coloring $p$ that refines $\pi$.*

This means GWL with any renep function, should it reach a coarsest stable coloring, will reach this unique coarsest stable coloring. It remains to show that GWL will reach a coarsest stable coloring.

**Theorem 6.** *For all $G$ the GWL refinement using any renep function will find the unique coarsest stable coloring of $V(G)$.*

*Proof.* Let $\pi_{\mathcal{T}} = \{p_1, \ldots, p_n\}$ be the stable coloring obtained from GWL on the initial coloring $\pi_0$. Assume there exists another stable coloring $\pi' = \{p'_1, \ldots, p'_m\}$ with $\pi_{\mathcal{T}} \prec \pi' \preccurlyeq \pi_0$, so $m < n$. Then $\exists v, w \in V(G) : (v \approx_{\pi'} w \wedge v \not\approx_{\pi_{\mathcal{T}}} w)$ and since Condition 4 applies $n(v) \neq n(w)$, which contradicts the assumption that $\pi'$ is stable. $\qquad\square$

The original Weisfeiler-Leman refinement can be realized by using the renep function with $\Leftrightarrow$ instead of $\Rightarrow$ in Condition 4. This ensures that vertices get assigned the same color, iff they previously had the same color and their neighborhood color multisets do not differ. Since this procedure splits all colors, that can be split up, it is the fastest converging possible renep function (because only direct neighborhood is considered). A trivial upper bound for the maximum number of Weisfeiler-Leman iterations needed is $|V(G)| - 1$ and there are infinitely many graphs on which this number of iterations is required for convergence [23]. We obtain the same upper bound for GWL.

**Theorem 7.** *The maximum number of iterations needed to reach the stable coloring using GWL refinement is $|V(G)| - 1$.*

*Proof.* The function we consider is a renep function. It follows that, prior to reaching the stable coloring, at least one color is split into at least two new colors in every iteration. Since vertices that had different colors at any step will also have different colors in the following iterations, the number of colors increases in every step. Hence, after at most $|V(G)| - 1$ steps, each vertex has a unique color, which is a stable coloring. $\qquad\square$

**Sequential Weisfeiler-Leman.** For optimizing the running time of the Weisfeiler-Leman algorithm, sequential refinement strategies have been proposed [24–26], which lead to the same stable coloring as the original WL. Our presentation follows Berkholz et al. [24], who provide implementation details and a thorough complexity analysis. Sequential WL manages a stack containing the colors that still have to be processed. All initial colors are added to this stack. In each step, the next color $c$ from the stack is used to refine the current coloring $\pi$ (and generate a new coloring $\pi'$) using the following update strategy: $\forall v, w \in V(G) \colon v \approx_{\pi'} w \Leftrightarrow \big|\{\!\!\{x \mid x \in N(v) \wedge \pi(x) = c\}\!\!\}\big| = \big|\{\!\!\{x \mid x \in N(w) \wedge \pi(x) = c\}\!\!\}\big| \wedge v \approx_{\pi} w$. Note that $\pi' \prec \pi$ is not guaranteed. For colors that are split, all new colors are added to the stack with exception of the largest color class. This is shown to be sufficient for generating the coarsest stable coloring [24].

Sequential Weisfeiler-Leman can be realized by our GWL with the restriction, that in sequential WL, some refinement operations might not produce strict refinements. We need to skip these in our approach (since renep functions have to produce strict refinements as long as the coloring is not stable). The renep function has to fulfill $\forall v, w \in V(G) \colon v \approx_{\pi_{\mathcal{T}_{i+1}}} w \Leftrightarrow \big|\{\!\!\{x \mid x \in N(v) \wedge \pi_{\mathcal{T}_i}(x) = c\}\!\!\}\big| = \big|\{\!\!\{x \mid x \in N(w) \wedge \pi_{\mathcal{T}_i}(x) = c\}\!\!\}\big| \wedge v \approx_{\pi_{\mathcal{T}_i}} w$, where $c$ is the next color in the stack that produces a strict refinement.

## 4.2 Running Time

The running time of the gradual Weisfeiler-Leman refinement depends on the cost of the update function used.

**Theorem 8.** *The running time for the gradual Weisfeiler-Leman refinement is $O(h \cdot t_u(|V(G)|))$, where $h$ is the number of iterations and $t_u(n)$ is the time needed to compute the renep function for $n$ elements.*

The update function used in the original Weisfeiler-Leman refinement can be computed in time $O(|V(G)| + |E(G)|)$ in the worst-case by sorting the neighbor color multisets using bucket sort [2].

## 4.3 Discussion of Suitable Update Functions

The update function of the original Weisfeiler-Leman refinement provides a fast way to reach the stable coloring, but in machine learning tasks a more fine grained vertex similarity might be desirable. A suitable update function restricts the number of new colors to a manageable amount, while still fulfilling the requirements of a renep function. Clustering the neighborhood multisets of the vertices, and letting the clusters imply the new colors, is an intuitive way to restrict the number of colors per iteration and assign similar neighborhoods the same new color. We discuss how to realize a renep function using clustering.

Whether two vertices, that currently have the same color, will be assigned the same color in the next step, depends on two factors: If they have the same neighbor color multiset, they have to remain in one color group. If their neighbor color multisets differ, however, the renep function can decide to either separate them or not (provided any new colors are generated to fulfill Condition 3). We propose clustering the neighbor color multisets separately for each old color and let the clusters imply new colors. If a clustering function guarantees to produce at least two clusters for inputs with at least two distinct objects, we obtain a renep function.

Although various clustering algorithms are available, we identified $k$-means as a convenient choice because of its efficiency and controllability of the number of clusters. In order to apply $k$-means to multisets of colors, we represent them as (sparse) vectors, where each entry counts the number of neighbors with a specific color. The above method using $k$-means clustering with $k > 1$ satisfies the requirements of a renep function. Of course, if the number of elements to cluster is less than or equal to $k$, the clustering can be omitted and each element can be assigned its own cluster. The number of clusters in iteration $i$ is bounded by $|L| \cdot k^i$, since each color can split into at most $k$ new colors in each iteration and the initial coloring has at most $|L|$ colors.

# 5 Applications

The gradual Weisfeiler-Leman refinement provides a more fine-grained approach to capture vertex similarity, where two vertices are considered more similar, the longer it takes until they get assigned different colors. This makes the approach applicable not only to vertex classification, but also in graph kernels and as a vertex similarity measure for graph matching. We further describe these possible applications in the following and evaluate them against the state-of-the-art methods in Section 6.

**Graph Kernels.** The idea of the GWL subtree kernel is essentially the same as the Weisfeiler-Leman subtree kernel [2], but instead of using the original Weisfeiler-Leman algorithm, the GWL algorithm is used to generate the features. We use the definition given in Equation (1) replacing the Weisfeiler-Leman colorings with the coloring from the GWL algorithm. The Weisfeiler-Leman optimal assignment kernel [5] is obtained from an optimal assignment between the vertices of two graphs regarding a vertex similarity defined by a color hierarchy. We replace the Weisfeiler-Leman color hierarchy used originally by the one from our gradual refinement. We evaluate the performance of our newly proposed kernels in Section 6.

**Tree Metrics for Approximating the Graph Edit Distance.** The graph edit distance, a general distance measure for graphs, is usually approximated due to its complexity. Many approximations find an optimal assignment between the vertices of the two graphs and derive a (sub-optimal) edit path from this assignment. Naturally, the cost of such an edit path is an upper bound for the graph edit distance. An optimal assignment can be computed in linear time, if the underlying cost function is a tree metric [27], which means an approximation of the graph edit distance can be found in linear time, given a tree metric on the vertices. The color hierarchy, see Figure 1, produced by the original Weisfeiler-Leman refinement, as well as our gradual variant, can be interpreted as such a tree metric. Lin [27] approximated the graph edit distance as described above. We can again replace the original color hierarchy by the one produced by our gradual Weisfeiler-Leman refinement. Appendix I includes a more detailed explanation on how to approximate the graph edit distance using assignments. We evaluate this approximation of the graph edit distance regarding its accuracy in $k$nn-classification against the state-of-the-art and the original approach.

# 6 Experimental Evaluation

We evaluate the proposed approach regarding its applicability in graph kernels, as the gradual Weisfeiler-Leman subtree kernel (GWL) and the gradual Weisfeiler-Leman optimal assignment kernel (GWLOA), as well as its usefulness as a tree metric for approximating the graph edit distance. Specifically, we address the following research questions:

**Q1** Can our kernels compete with state-of-the-art methods regarding classification accuracy on real-world and synthetic datasets?

**Q2** Which refinement speed is appropriate and are there dataset-specific differences?

**Q3** How do our kernels compare to the state-of-the-art methods in terms of running time?

**Q4** Is the vertex similarity obtained from GWL refinement suitable for approximating the graph edit distance and solving learning problems with it?

We compare to the Weisfeiler-Leman subtree kernel (WLST) [2], the Weisfeiler-Leman optimal assignment kernel (WLOA) [5], as well as the approximation of the relaxed Weisfeiler-Leman subtree kernel (RWL*) [15], and the deep Weisfeiler-Leman kernel (DWL) [13]. We do not compare to [4], since the kernel showed results similar to the WLOA kernel. We compare the graph edit distance approximation using our tree metric GWLT to the original approach Lin [27] and state-of-the-art method BGM [28]. Our implementation is publicly available at `https://github.com/frareba/GradualWeisfeilerLeman`.

## 6.1 Setup

As discussed in Section 4.3 we used $k$-means clustering in our new approach. If for any color less than $k$ different vectors were present in the clustering step, each distinct vector got its own cluster. We implemented our GWL, GWLOA and also the original WLST and WLOA in Java. We used the

**Table 1:** Average classification accuracy and standard deviation (highest accuracies marked in **bold**).

| Kernel | PTC_FM | KKI | EGO-1 | EGO-2 | EGO-3 | EGO-4 |
|---|---|---|---|---|---|---|
| WLST | 64.16 $\pm$1.30 | 49.97 $\pm$2.88 | 51.30 $\pm$2.42 | 57.15 $\pm$1.61 | 56.15 $\pm$1.67 | 53.40 $\pm$1.77 |
| DWL | 64.18 $\pm$1.46 | 50.93 $\pm$2.87 | 55.80 $\pm$1.35 | 56.50 $\pm$1.64 | 55.90 $\pm$1.64 | 53.25 $\pm$2.81 |
| RWL* | 62.43 $\pm$1.46 | 46.54 $\pm$4.03 | 65.60 $\pm$2.74 | 70.20 $\pm$1.36 | **67.60** $\pm$1.07 | 74.25 $\pm$2.12 |
| WLOA | 62.34 $\pm$1.39 | 48.72 $\pm$4.05 | 55.95 $\pm$1.11 | 60.30 $\pm$2.00 | 54.25 $\pm$1.35 | 52.30 $\pm$2.29 |
| **GWL** | 62.61 $\pm$1.94 | **57.79** $\pm$3.95 | 67.95 $\pm$2.05 | **73.65** $\pm$1.86 | 65.45 $\pm$1.88 | **77.45** $\pm$1.97 |
| **GWLOA** | **64.58** $\pm$1.77 | 47.47 $\pm$2.41 | **69.80** $\pm$1.65 | 72.40 $\pm$2.52 | 67.45 $\pm$1.69 | 75.35 $\pm$1.67 |

| | COLLAB | DD | IMDB-B | MSRC_9 | NCI1 | REDDIT-B |
|---|---|---|---|---|---|---|
| WLST | 78.98 $\pm$0.22 | 79.00 $\pm$0.52 | 72.01 $\pm$0.80 | 90.13 $\pm$0.75 | 85.96 $\pm$0.18 | 80.81 $\pm$0.52 |
| DWL | 78.93 $\pm$0.18 | 78.92 $\pm$0.40 | 72.36 $\pm$0.56 | 90.50 $\pm$0.76 | 85.68 $\pm$0.18 | 80.83 $\pm$0.40 |
| RWL* | 77.94 $\pm$0.38 | 77.52 $\pm$0.65 | 72.96 $\pm$0.86 | 88.86 $\pm$0.89 | 79.45 $\pm$0.32 | 77.69 $\pm$0.31 |
| WLOA | 80.81 $\pm$0.22 | **79.44** $\pm$0.31 | 72.60 $\pm$0.89 | 90.68 $\pm$0.92 | **86.29** $\pm$0.13 | 89.40 $\pm$0.14 |
| **GWL** | 80.62 $\pm$0.33 | 79.00 $\pm$0.81 | **73.66** $\pm$1.25 | 88.32 $\pm$1.20 | 85.33 $\pm$0.35 | 86.46 $\pm$0.35 |
| **GWLOA** | **81.30** $\pm$0.29 | 78.49 $\pm$0.57 | 72.88 $\pm$0.79 | **91.27** $\pm$1.06 | 85.36 $\pm$0.36 | **89.98** $\pm$0.34 |

RWL* and DWL Python implementations provided by the authors. Note that in contrast to the other approaches, the RWL* implementation uses multi-threading.

For evaluation, we used the $C$-SVM implementation LIBSVM [29] and report average classification accuracies obtained by 10-fold nested cross-validation repeated 10 times with random fold assignments. The parameters of the SVM and the kernels were optimized by the inner cross-validation on the current training fold using grid search. We chose $C \in \{10^{-3}, 10^{-2}, \dots, 10^3\}$ for the SVM, $h \in \{0, \dots, 10\}$ for WLST, WLOA, GWL and GWLOA and k-means with $k \in \{2, 4, 8, 16\}$. For RWL* we used $h \in \{1, \dots, 4\}$ and default values for the other parameters. In DWL the window size $w$ and dimension $d$ were set to 25, since this generally worked best out of the combinations from $d, w \in \{5, 25, 50\}$ and no defaults were provided. We used the default settings for the other parameters and again $h \in \{1, \dots, 10\}$. The running time experiments were conducted on an Intel Xeon Gold 6130 machine at 2.1 GHz with 96 GB RAM. For evaluating the approximation of the graph edit distance, we compare the 1-nn classification accuracy and used the Java implementation of Lin provided by the authors and implemented our approach GWLT, as well as BGM, also in Java for a fair comparison.

**Extension to Edge Labels.** The original Weisfeiler-Leman algorithm can be extended to respect edge labels by updating the colors according to $c_{i+1}(v) = z(c_i(v), \{\!\{(l(u, v), c_i(u)) \mid u \in N(v)\}\!\})$. All kernels used in the comparison use a similar strategy to incorporate edge labels if present.

**Datasets.** We used several real-world datasets from the TUDataset [30] and the *EGO-Nets* datasets [15] for our experiments. See Appendix B and F for an overview of the datasets, as well as additional synthetic datasets and corresponding results. We selected these datasets as they cover a wide range of applications, consisting of both molecule datasets and graphs derived from social networks. See Appendix C for the number of Weisfeiler-Leman iterations needed to reach the stable coloring for each dataset.

## 6.2 Results

In the following, we present the classification accuracies and running times of the different kernels. We investigate the parameter selection for our algorithm and discuss the application of our approach for approximating the graph edit distance.

**Q1: Classification Accuracy.** Table 1 shows the classification accuracy of the different kernels. While on some datasets our new approaches do not outcompete all state-of-the-art methods, they are more accurate in most cases, in some cases even with a large margin to the second-placed (for example on *KKI*, *EGO-1* or *EGO-4*). While RWL* is better than our approaches on some datasets, the running time of this method is much higher, cf. Q3. WLOA also produces very good results on many datasets, but cannot compete on the *EGO-Nets* and synthetic datasets (see Appendix F). For

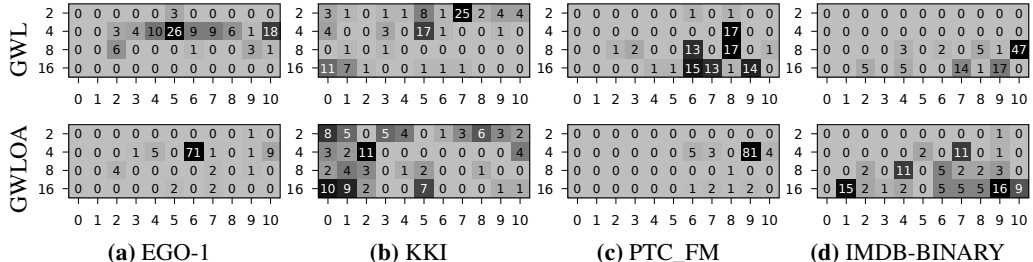

**Figure 3:** Number of times a parameter combination of GWL and GWLOA was selected from $k \in \{2, 4, 8, 16\}$ and $h \in \{0, \ldots, 10\}$ based on the accuracy achieved on the test set.

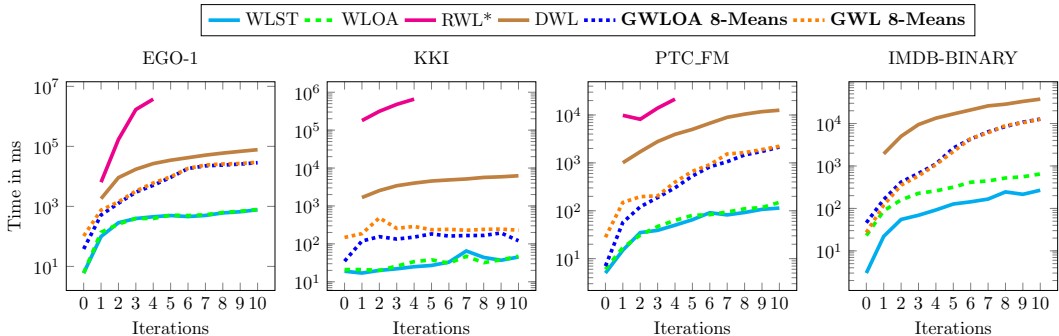

**Figure 4:** Running time in milliseconds for computing the feature vectors for all graphs of a dataset using the different methods. Note that RWL* uses multi-threading, while the other methods do not. Missing values for RWL* and DWL in the larger datasets are due to timeout.

molecular graphs (*PTC_FM*, *NCI*) we see no significant improvements, which can be explained by their small degree and sensitivity of molecular properties to small changes. Overall, our method provides the highest accuracy on 9 of 12 datasets and is close to the best accuracy for the others.

**Q2: Parameter Selection.** For GWL and GWLOA two parameters have to be chosen: The number of iterations $h$ and the number $k$ of clusters in $k$-means. We investigate which choices lead to the best classification accuracy. Figure 3 shows the number of times, a specific parameter combination was selected as it provided the best accuracy for the test set. Here, we only show the parameter selection for some of the datasets. The results for the other datasets, as well as the parameter selection for WLST and WLOA, can be found in Appendix D. We can see that for GWL and most datasets the best $k$ is in $\{2, 4, 8\}$ and on those datasets classification accuracy of GWL exceeds that of WLST. On datasets on which GWL performed worse than WLST, the best choice for parameters is not clear and it seems like a larger $k$ might be beneficial for improving the classification accuracy. Similar tendencies can be observed for GWLOA.

**Q3: Running Time.** Figure 4 shows the time needed for computing the feature vectors using the different kernels (for results on the other datasets and the influence of the parameter $k$ on running time see Appendix E and G). RWL* and DWL are much slower than the other kernels, while only RWL* leads to minor improvements in classification accuracy on few datasets. While our approach is only slightly slower than WLST/WLOA, it yields great improvements on the classification accuracy on most datasets, cf. Q1.

**Q4: Learning with Approximated Graph Edit Distance.** Table 2 compares the 1-nn classification accuracy of our approach when approximating the graph edit distance to the original [27] and another state-of-the-art method based on bipartite graph matching [28]. Our approach clearly outcompetes both methods on all datasets.

We investigate the approximation quality for similar graphs (which are important for $k$-nn classification) further by comparing the approximation of Lin [27] to our method GWLT directly, see Figure 5.

**Table 2:** Average classification accuracy and standard deviation (highest accuracies marked in **bold**).

| Method | PTC_FM | MSRC_9 | KKI | EGO-1 | EGO-2 | EGO-3 | EGO-4 |
|---|---|---|---|---|---|---|---|
| BGM | 60.14 ±1.50 | 72.13 ±1.28 | 43.89 ±1.27 | 44.75 ±1.05 | 42.05 ±1.25 | out of time | out of time |
| Lin | 62.38 ±1.08 | 81.36 ±0.64 | **55.18** ±2.44 | 40.40 ±1.17 | 31.65 ±1.07 | 26.60 ±0.94 | 36.55 ±1.72 |
| **GWLT** | **63.19** ±0.11 | **85.97** ±0.59 | **55.18** ±2.44 | **56.20** ±1.42 | **47.90** ±1.28 | **36.40** ±1.04 | **47.90** ±1.32 |

| | | |
|---|---|---|
| **(a)** PTC_FM | **(b)** MSRC_9 | **(c)** KKI |
| **(d)** EGO-1 | **(e)** EGO-2 | **(f)** EGO-3 |

**Figure 5:** Approximation quality of GWLT compared to Lin [27]. Marks below the gray diagonal indicate that GWLT had a better approximation, while marks above indicate the same for Lin.

Only graph pairs for which at least one of the methods returned a distance below a cutoff threshold are depicted to emphasize the important case of highly similar graphs. The results clearly show that GWLT provides more accurate approximations, in particular, for the *EGO-Nets*. This finding and the accuracy results discussed in Q1 indicate that on these datasets methods benefit from the more fine-grained vertex similarity provided by GWL.

## 7   Conclusions

We proposed a general framework for iterative vertex refinement generalizing the popular Weisfeiler-Leman algorithm and discussed connections to other vertex refinement strategies. Based on this, we proposed two new graph kernels and showed that they outperform the original Weisfeiler-Leman subtree kernel and similar state-of-the-art approaches in terms of classification accuracy in almost all cases, while keeping the running time much lower than comparable methods. We also investigated the application of our method to approximating the graph edit distance, where we again outperformed the state-of-the-art methods.

In further research, other renep functions can be explored, for example, by using different clustering strategies or developing new concepts for inexact neighborhood comparison. Moreover, we will systematically relate our approach to graph neural networks and investigate whether similar ideas can be incorporated into their neighborhood aggregation step and to what extent common architectures and optimization methods are capable of learning certain renep functions.

## Acknowledgements

This work has been supported by the Vienna Science and Technology Fund (WWTF) through project VRG19-009.

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

**Table 3:** Datasets with discrete vertex and edge labels and their statistics [30]. The *EGO-Nets* datasets [15] are unlabeled.

| Name | \|**Graphs**\| | \|**Classes**\| | **avg** $\|V\|$ | **avg** $\|E\|$ | $\|L_V\|$ | $\|L_E\|$ |
|------|-------|---------|--------|--------|-------|-------|
| *KKI* | 83 | 2 | 26.96 | 48.42 | 190 | – |
| *PTC_FM* | 349 | 2 | 14.11 | 14.48 | 18 | 4 |
| *COLLAB* | 5000 | 3 | 74.49 | 2457.78 | – | – |
| *DD* | 1178 | 2 | 284.32 | 715.66 | 82 | – |
| *IMDB-BINARY* | 1000 | 2 | 19.77 | 96.53 | – | – |
| *MSRC_9* | 221 | 2 | 40.58 | 97.94 | 10 | – |
| *NCI1* | 4110 | 2 | 29.87 | 32.30 | 37 | – |
| *REDDIT-BINARY* | 2000 | 2 | 429.63 | 497.75 | – | – |
| *EGO-1* | 200 | 4 | 138.97 | 593.53 | – | – |
| *EGO-2* | 200 | 4 | 178.55 | 1444.86 | – | – |
| *EGO-3* | 200 | 4 | 220.01 | 2613.49 | – | – |
| *EGO-4* | 200 | 4 | 259.78 | 4135.80 | – | – |

## A  Pseudocode

Algorithm 1 shows the procedure of gradual Weisfeiler-Leman refinement. We start with an initial coloring (either uniform or based on the vertex labels) and then iteratively refine these colors using a renep function, $h$ times.

---

**Algorithm 1** Gradual Weisfeiler-Leman Refinement.

---

```
 1: procedure GWL(G, h)                              ▷ h is number of iterations
 2:     for all v ∈ V(G) do                          ▷ Initial coloring
 3:         c₀(v) ← μ(v)
 4:     initialize T₀ as described in Section 3
 5:     for i ← 1, i ≤ h, i ← i + 1 do
 6:         𝒯ᵢ ← f(G, 𝒯ᵢ₋₁)                          ▷ Compute new colors
 7:         for all v ∈ V(G) do                      ▷ Assign new colors
 8:             cᵢ(v) ← π_{𝒯ᵢ}(v)
```

---

## B  Datasets

We used several real-world datasets from the TUDataset [30], the *EGO-Nets* datasets [15], as well as synthetic datasets for our experiments. See Table 3 for an overview of the real-world datasets. We selected these datasets as they cover a wide range of applications, consisting of both molecule datasets and graphs derived from social networks.

The synthetic datasets were generated using the block graph generation method [15]. We generated 9 synthetic datasets with two classes and 200 graphs in each class. For each dataset we first generated two seed graphs (one per class) with 16 vertices, that both are constructed from a tree by appending a single edge, so that their sets of vertex degrees are equal. For the dataset graphs each vertex of the seed graph was replaced by 8 vertices. Vertices generated from the same seed vertex, as well as from adjacent vertices are connected with probability $p$. $m$ noise edges are then added randomly. We investigated the two cases $p = 1.0$ and $m \in \{0, 10, 20, 50, 100\}$, and $p \in \{1.0, 0.8, 0.6, 0.4, 0.2\}$ and $m = 0$. We denote the datasets by *S_p_m*.

## C  Number of Weisfeiler-Leman Iterations

We investigate the number of WL iterations needed to reach the stable coloring in the various datasets. On the datasets with $-$ entries (see Table 4) our algorithm, that checked whether the stable coloring is reached, did not finish in a reasonable time due to the size of the datasets/graphs. It can be seen

**Table 4:** Number of iterations needed to reach the stable coloring.

| Dataset | KKI | PTC_FM | COLLAB | DD | IMDB-B | MSRC_9 |
|---------|-----|--------|--------|-----|--------|--------|
| WL | 3 | 13 | – | – | 3 | 3 |

| Dataset | NCI1 | REDDIT-B | EGO-1 | EGO-2 | EGO-3 | EGO-4 |
|---------|------|----------|-------|-------|-------|-------|
| WL | 39 | – | 5 | 4 | 4 | 5 |

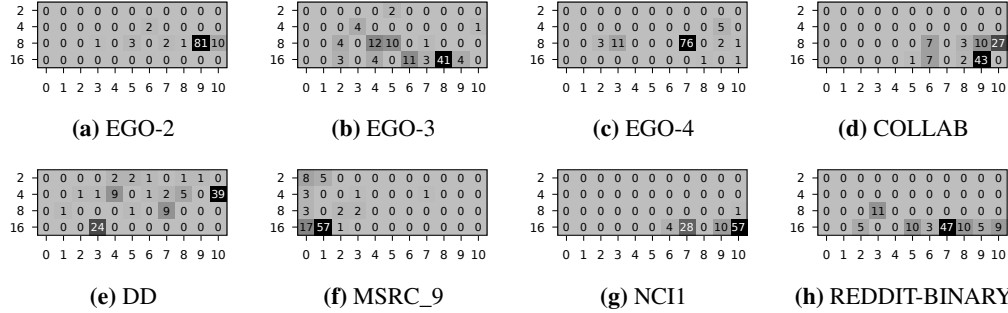

**(a)** EGO-2  **(b)** EGO-3  **(c)** EGO-4  **(d)** COLLAB

**(e)** DD  **(f)** MSRC_9  **(g)** NCI1  **(h)** REDDIT-BINARY

**Figure 6:** With $k \in \{2, 4, 8, 16\}$ and $h \in \{0, \dots, 10\}$, we show the number of times a specific parameter combination for GWL was selected as it provided the best accuracy for the test set.

that on most datasets the number of iterations needed to reach the stable coloring is very low. This means that after a few iterations we do not gain any new information when using for example the traditional WLST kernel.

## D Parameter Selection - Further Results

Figures 6 and 7 show the parameter selection for the remaining datasets for GWL and GWLOA. In most datasets the choice is restricted to two or three values. For some of the datasets, the best choice seems to include $k = 16$. This might indicate that a larger $k$ could be beneficial for increasing the accuracy.

Figure 8 shows the parameter selection for WLST and WLOA. There is only one parameter, the number of WL iterations, for both kernels. We can see that indeed on most datasets, only few iterations are needed to gain the best possible accuracy. On datasets such as *EGO-2*, *EGO-4* or *NCI1*, however, this is not the case. For *NCI1* we can assume that we still gain information through more iterations, since we have not yet reached the stable coloring. For the *EGO*-datasets, this is surprising, since the stable coloring is reached after 4 (5) iterations. On the other hand, the classification accuracy reached is still not good.

**(a)** EGO-2  **(b)** EGO-3  **(c)** EGO-4  **(d)** COLLAB

**(e)** DD  **(f)** MSRC_9  **(g)** NCI1  **(h)** REDDIT-BINARY

**Figure 7:** With $k \in \{2, 4, 8, 16\}$ and $h \in \{0, \dots, 10\}$, we show the number of times a specific parameter combination for GWLOA was selected as it provided the best accuracy for the test set.

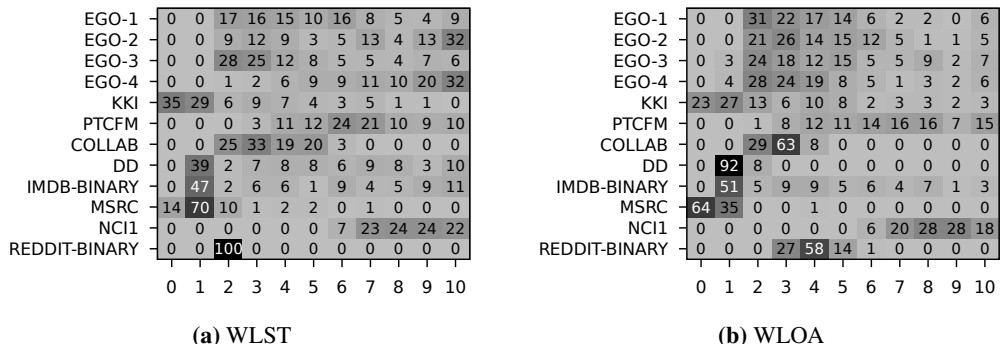

**(a)** WLST                    **(b)** WLOA

**Figure 8:** With $h \in \{0, \ldots, 10\}$, we show the number of times a specific parameter combination for WLST and WLOA was selected as it provided the best accuracy for the test set.

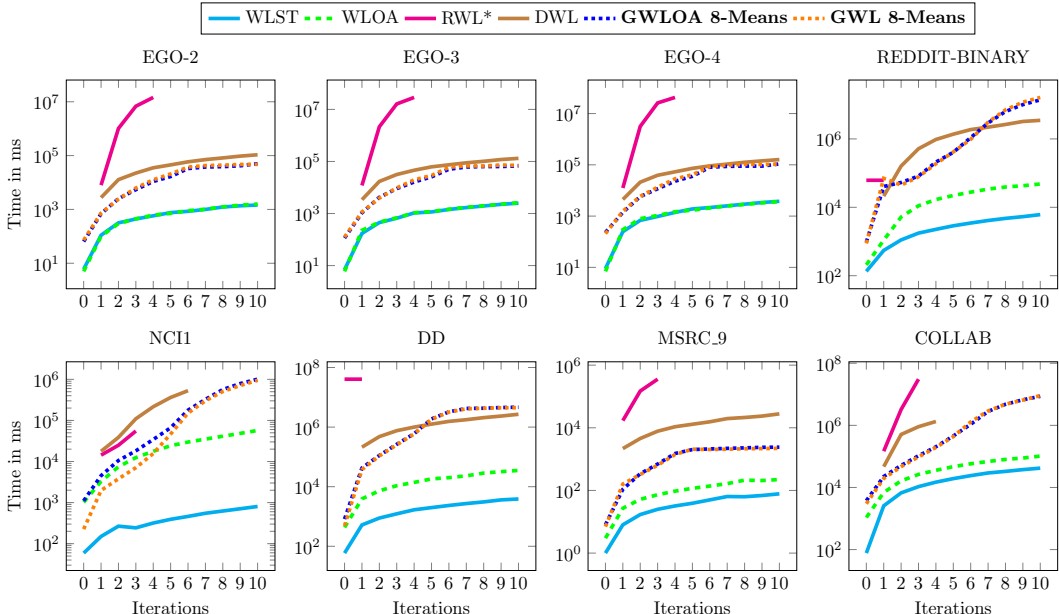

**Figure 9:** Running time in milliseconds for computing the feature vectors using the different methods. Note that RWL* uses multi-threading, while the other methods do not. Missing values for RWL* and DWL in the larger datasets are due to timeout.

## E    Running Time - Further Results

Figure 9 shows the running time results for the remaining datasets. While the runtime of our approach exceeds that of DWL on some of the larger datasets, it enhances the classification accuracy a lot.

## F    Results on Synthetic Datasets

Table 5 shows the classification accuracy of the different methods on the synthetic datasets (generated as described in Appendix B), with the best accuracy for each dataset being marked in bold. We can see, while all kernels can perfectly learn on the datasets without noise, neither WLST, WLOA nor DWL can manage the noise included in the other datasets, having worse accuracy with increasing noise. While the decrease in accuracy with decreasing the edge probability is slightly worse than that of RWL*, our approach has a much lower running time.

**Table 5:** Average classification accuracy and standard deviation on the synthetic datasets.

| Kernel | $S\_1\_0$ | $S\_1\_10$ | $S\_1\_20$ | $S\_1\_50$ | $S\_1\_100$ | $S\_0.8\_0$ | $S\_0.6\_0$ | $S\_0.4\_0$ | $S\_0.2\_0$ |
|---|---|---|---|---|---|---|---|---|---|
| WLST | **100.00**$_{\pm 0.00}$ | 98.68 $_{\pm 0.34}$ | 61.93 $_{\pm 1.08}$ | 54.55 $_{\pm 0.68}$ | 49.78 $_{\pm 1.18}$ | 50.65 $_{\pm 1.57}$ | 48.10 $_{\pm 1.10}$ | 51.98 $_{\pm 1.40}$ | 42.65 $_{\pm 1.80}$ |
| DWL | **100.00**$_{\pm 0.00}$ | 98.70 $_{\pm 0.31}$ | 62.10 $_{\pm 0.98}$ | 43.83 $_{\pm 1.66}$ | 51.40 $_{\pm 0.94}$ | 49.80 $_{\pm 2.01}$ | 46.88 $_{\pm 1.89}$ | 49.05 $_{\pm 1.98}$ | 42.85 $_{\pm 1.98}$ |
| RWL* | **100.00**$_{\pm 0.00}$ | **100.00**$_{\pm 0.00}$ | **100.00**$_{\pm 0.00}$ | out of time | **100.00**$_{\pm 0.00}$ | **100.00**$_{\pm 0.00}$ | **99.35** $_{\pm 0.17}$ | **81.93** $_{\pm 1.00}$ | **56.33** $_{\pm 2.48}$ |
| WLOA | **100.00**$_{\pm 0.00}$ | 97.65 $_{\pm 0.44}$ | 60.85 $_{\pm 1.65}$ | 47.50 $_{\pm 1.83}$ | 50.23 $_{\pm 1.24}$ | 49.45 $_{\pm 1.47}$ | 48.08 $_{\pm 1.92}$ | 43.23 $_{\pm 1.21}$ | 50.53 $_{\pm 1.92}$ |
| **GWL** | **100.00**$_{\pm 0.00}$ | **100.00**$_{\pm 0.00}$ | **100.00**$_{\pm 0.00}$ | **100.00**$_{\pm 0.00}$ | **100.00**$_{\pm 0.00}$ | **100.00**$_{\pm 0.00}$ | 92.20 $_{\pm 0.97}$ | 72.53 $_{\pm 1.78}$ | 53.58 $_{\pm 1.71}$ |
| **GWLOA** | **100.00**$_{\pm 0.00}$ | **100.00**$_{\pm 0.00}$ | **100.00**$_{\pm 0.00}$ | **100.00**$_{\pm 0.00}$ | **100.00**$_{\pm 0.00}$ | **100.00**$_{\pm 0.00}$ | 94.95 $_{\pm 0.59}$ | 72.03 $_{\pm 1.66}$ | 50.90 $_{\pm 2.63}$ |

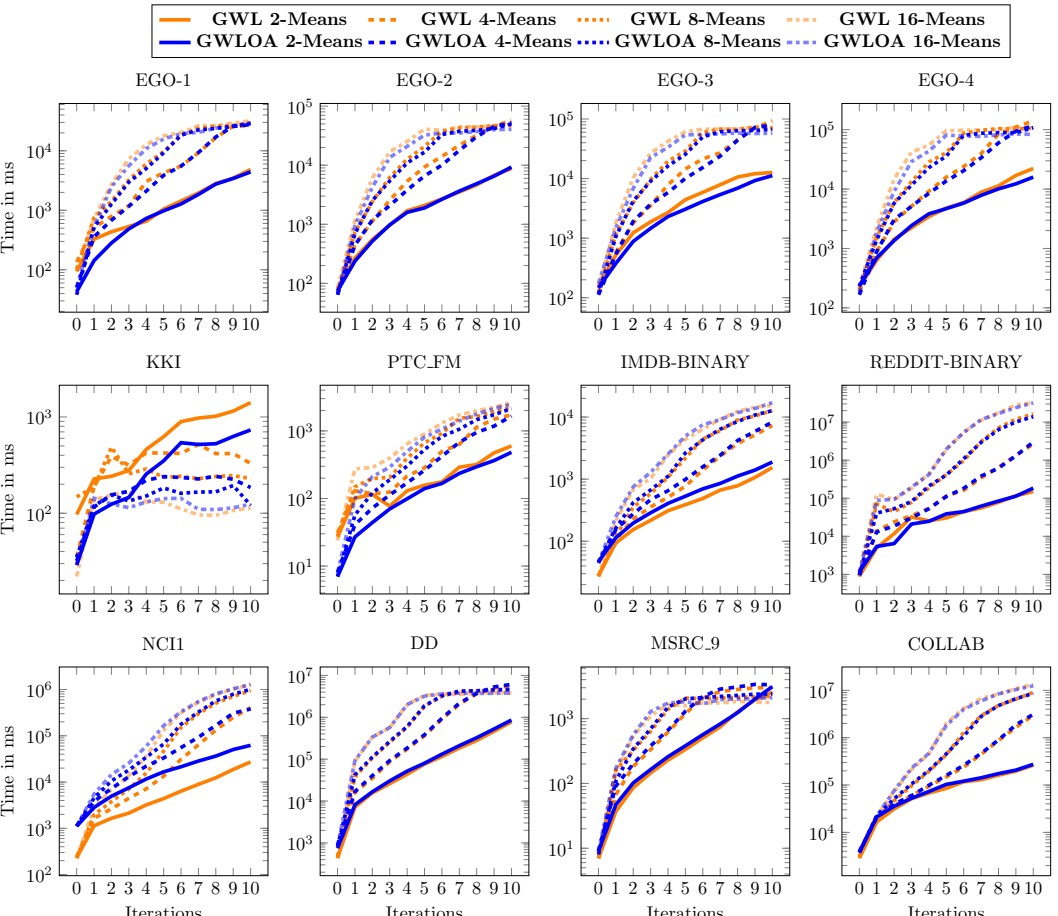

**Figure 10:** Running time in milliseconds for computing the feature vectors using the different values for parameter $k$ on our newly proposed methods.

## G  Influence of Parameter $k$ on Running Time

We investigate which effect the choice of $k$ has on the running time. Figure 10 shows the time needed for computing the feature vectors using our kernels with $k \in \{2, 4, 8, 16\}$. The difference in running time between GWL and GWLOA is only marginal on most datasets, only on *KKI* and *NCI1* a larger difference can be seen. As expected, the running time of both kernels increases with increasing $k$. Interestingly, for larger $k$, the running time does not increase much anymore, after a certain number of iterations, this might be because the stable coloring was reached by then.

## H  Results on Larger Datasets

GWL allows to generate explicit sparse feature vectors just as WLST. Therefore, these kernels can be used with a linear SVM, which is more efficient than a kernel SVM and makes the application to larger datasets feasible. We performed additional experiments with these kernels on larger synthetic

**Table 6:** Parameters used for generating the larger datasets.

| Dataset | $p$ | $m$ | \|**Graphs**\| | $b$ | $r$ |
|---------|-----|-----|------------|-----|-----|
| *L1* | 1 | 200 | 10000 | 25 | 10 |
| *L2* | 1 | 100 | 20000 | 25 | 10 |
| *L3* | 1 | 200 | 20000 | 10 | 15 |
| *L4* | 1 | 400 | 20000 | 25 | 10 |

**Table 7:** Results on larger datasets with running time in seconds and $|f|$ being the number of features (for WLST we only give the order of magnitude).

| Kernel | *L1* | | | *L2* | | | *L3* | | | *L4* | | |
|--------|------|-----|------|------|------|------|------|------|------|------|------|------|
| | time | acc | $|f|$ | time | acc | $|f|$ | time | acc | $|f|$ | time | acc | $|f|$ |
| WLST | 25 | 50.92 | $10^6$ | 59 | 84.57 | $10^6$ | 45 | 49.91 | $10^6$ | 124 | 49.65 | $10^6$ |
| GWL | 208 | 99.98 | 61 | 237 | 100.0 | 15 | 103 | 100.0 | 7 | 137 | 99.98 | 7 |

datasets using the linear SVM implementation LIBLINEAR [31]. The datasets were generated using the same method as described in Appendix B, but with different values for the number of vertices in the seed graphs, $b$, and number of vertices, each vertex in the seed graph is replaced by, $r$ (see Table 6 for the values of the parameters). The experimental setup used in these experiments was as follows: We split the dataset randomly into a training, validation and test set and chose the parameter $C \in \{0.1, 1, 10\}$. We used $h \in \{1, \ldots, 5\}$ for both kernels and set $k = 2$ for all datasets.

In Table 7 we report the running time and the number of features $f$ obtained for the choice of $h$ that gave the best classification accuracy (for GWL this choice was $h = 2$ for *L4* and *L5*, $h = 3$ for *L2* and $h = 5$ for *L1*, for WLST it was $h = 2$ for all datasets except *L4*, on which it was $h = 5$). We noticed that running the SVM with the feature vectors generated by WLST took much more time than GWL, which can be explained by the number of features each algorithm produced: For GWL the number is restricted by the choice of parameters, but for WLST the number of features is only restricted by the number of vertices in the dataset. In WLST the number of features exceeded $10^6$ for $h \geq 2$. For a small increase in running time in generating the feature vectors, GWL provides not only a much better classification accuracy than WLST, it also generates less features, but more meaningful ones.

# I    Approximating the Graph Edit Distance using Optimal Assignments

The graph edit distance (GED), a commonly used distance measure for graphs, is defined as the cost of transforming one graph into the other using edit operations, i.e. deleting or inserting an isolated vertex or an edge, or relabeling any of the two. An edit path between $G$ and $H$ is a sequence $(e_1, e_2, \ldots, e_k)$ of edit operations that transforms $G$ into $H$. This means, that applying all operations in the edit path to $G$, yields a graph $G'$ that is isomorphic to $H$.

Edit operations have non-negative costs assigned to them by a cost function $c$ and the GED of two graphs is defined as the cost of a cheapest edit path between them:

$$\text{GED}(G, H) = \min \left\{ \sum_{i=1}^{k} c(e_i) \mid (e_1, \ldots, e_k) \in \Upsilon(G, H) \right\},$$

where $\Upsilon(G, H)$ denotes the set of all possible edit paths from $G$ to $H$.

Since computing the GED is NP-hard [32], it is often approximated; often an optimal assignment between the vertices of the graphs is computed and a (suboptimal) edit path is derived from this assignment [28]. Figure 11 shows two graphs, an assignment between their vertices indicated by matching numbers, and a corresponding edit path.

If the cost function is a tree metric, such an assignment can be computed in linear time [27] (as opposed to cubic runtime for arbitrary cost functions). The color hierarchy produced by for example GWLT can be interpreted as such a tree metric, which means it can be used to find an optimal

**Figure 11:** Two graphs $G$ and $H$ with an assignment between their vertices (vertices with matching numbers are assigned to each other) and an edit path derived from this assignment.

assignment between the vertices and in turn approximate the graph edit distance. Given a tree $T$, representing a tree metric, an optimal assignment between two sets $A$ and $B$ can be computed as follows: Let $\phi$ be a function that associates all elements with their node in the tree (for GWLT this means, that we associate each vertex with its "newest" color). Then deconstruct $T$ fully (until there are no leaves left) by repeating these steps:

1. Pick a random leaf $l$.
2. Assign as many elements of $A$ associated with $l$ to elements of $B$ associated with $l$ as possible.
3. Re-associate remaining elements to the parent of $l$.
4. Delete $l$ from $T$.

The resulting assignment is optimal and can be used to derive an edit path as above.

