# OpenReview forum: "Gradual Weisfeiler-Leman: Slow and Steady Wins the Race"
_logconference.io/LOG/2022/Conference — LoG 2022 Poster_

### Official Review · Reviewer_3yZr · 2022-10-14

**Overall Score:** 8
**Confidence:** 5

**Review:**


The authors propose to 'slow down' the Weisfeiler Leman color refinement algorithm by replacing the injective label compression function (of the original algorithm) by a non-injective function with special properties.
The proposed refining, neighborhood preserving (renep) function class guarantees convergence to the stable coloring of Weisfeiler Leman but usually requires more iterations to reach the stable coloring.
The authors show that there are practically efficient and effective ways of defining such renep functions, which result in good predictive performance on graph classification benchmark tasks.

## Pros
- The paper is well written and easy to follow.
- The method is clearly presented
- the proposed method offers a solution to a problem that almost all WL-based learning methods face, namely the (practically too) fast convergence of WL to a stable coloring.
- the empirical results are convincing regarding speed and predictive performance (of the kernel variants)

## Cons
- There seems to be a small issue with the definition of color hierarchies (see below)
- Q4 should be rephrased, as it is currently not answered, in my opinion.
- the description of the evaluation is not detailed enough


## Verdict
The paper provides an interesting and practically efficient solution to a problem that many WL based algorithms face: The fast convergence to a stable coloring.
The proposed method can be used as a drop-in for WL label refinement and shows promising empirical results in both running time and predictive performance, compared to previous state of the art.

I recommend this paper to be accepted to LOG.
I believe that the negative points that I have listed can be fixed, as indicated below. I am confident that the authors can do this in the revision period, if they agree.


## Transductive Color Hierarchy Definition
It seems that your definition of a color hierarchy as containing identical colors in children if the associated vertex sets are equal makes this definition transductive. This results in a subtly broken connection to the definition of the traditional WL color refinement.

More specifically, consider a path of lenght six and a path of length four with a path of length three attached to the central node. [Check out my drawing](https://ibb.co/5TKmLmf).
If we consider the two graphs individually, then the two leaves of the path of length six will have the same color under Weisfeiler-Leman in $\pi^P_1, \pi^P_2, \pi^P_3$ (according to the definition in Line 102ff.
However, in the second graph, the leaves will have the same color under Weisfeiler-Leman in $\pi^G_1$ and $\pi^G_2$, but then split up in $\pi^G_3$ and hence have different colors, which results in the same WL unfolding tree having different colors in two different graphs, which does not go well with a (unspecified) function that assigns colors (globally?) based on multisets of labels.

This would be fine considering your comment in line 161--165. However, this note comes rather late.
Before we see this cautionary note, the definition of the WL kernel in Eq. (1) does assume two independent graphs $G_1, G_2$ and an injective function $h$ computing the color updates.
In particular, this function $h$ is not required to produce a color hierarchy with identical colors whenever the vertex sets are (globally) equal. As a result, I am confused.

May I suggest getting rid of the 'color compression' in lines 103-105? It seems they are not needed for the definition of a stable coloring (cf. l 118f) and are confusing (at least for me).


## Q4: Is the vertex similarity obtained from GWL refinement suitable for approximating the graph edit distance?

While this is an interesting question, it seems that the answer given for Q4 in the experimental section in fact not concerned with the approximation of the graph edit distance.
Would an exact computation of the GED result in the best kNN accuracy on the given learning tasks? Possible, but you cannot conclude from better predictive performance that you have a better approximation of the graph edit distance.
In fact, what you show is that the particular (dis-)similarity that you compute is better suited to solve a learning task than other methods, this is similar to the results shown in table 1, but for a different classifier and similarity variant.

I think that rephrasing Q4 accordingly solves this issue.

## Description of Evaluation
I suggest you add furter details on experimental evaluation in the supplementary. In particular, I assume that e.g. the results in Table 1 were obtained using the kernels in an SVM, but this is never mentioned. Do you do cross validation? Etc. Same holds for The experiments whose results are shown in Table 2. How many neighbors were used in the kNN classifier, ...?

## Minor Issues:
- l116: the definition of $h$ implicitly defines the colors to be natural numbers, which was never discussed before. Maybe you can specify this in line 88 already: The set $L$ contains categorical labels[, i.e. $L \subseteq \mathbb{N}_0$].
- l116/ eq. (1) $h$ is overloaded as injective function and the number of iterations in the WL subtree kernel.
- l47f: 'We discuss [a] possible realization '
- l147: V(G).v -> V(G): v
- l225: renap -> renep
- l341: 'Our approach clearly outcompetes both method[s]'

---

### Official Review · Reviewer_BiT4 · 2022-10-18

**Overall Score:** 5
**Confidence:** 4

**Review:**

Summary
--

The main contribution of this paper is a variant of the 1-dimensional Weisfeiler-Leman (WL)algorithm which does not use an injective relabeling function, but instead, some function that might also map nodes that have structurally similar neighborhoods to the same label. The authors propose two graph kernels that utilize the above variant of WL and evaluate them on graph classification datasets where they achive high levels of performance. They also show that the proposed algorithm can accurately approximate the graph edit distance.

Strong points
--
- This is an interesting paper which follows a line of work (including refs [5] and [14] from the paper) that aims to make the WL algorithm more attractive for machine learning applications. Indeed, WL was originally designed for testing isomorphism of graphs, while machine learning problems require finer node similarity measures. I also believe that the paper might inspire more research on this direction since this is a topic that has not been well investigated in the literature so far.

- The idea is novel. The idea is also relatively simple which a good thing in my opinion. Instead of directly splitting a color into multiple (pairwise orthogonal) colors, the algorithm gradually splits the color into these colors and thus nodes with structurally similar neighborhoods share the same colors for more iterations of the algorithm.

- Another strength of the proposed algorithm is that it does not incur any high additional computational cost in case simple update functions (such as $k$-means) are employed. This is of high importance since real-world graphs usually consist of a large number of nodes.


Weak points
--
- One of the major weak points of the paper is that the proposed kernels were evaluated only on small datasets. In fact, 2/3 of the considered datasets contain 1,000 graphs or less, while the largest dataset consists of 5,000 graphs in total. This is in contrast to most recent graph property prediction datasets which contain hundreds of thousands of graphs (see ogb datasets for instance). This is a fundamental limitation inherent to graph kernels which questions their applicability to several recently introduced tasks. I would suggest the authors leverage approximation algorithms to apply the proposed kernels to some larger dataset.

- Most recent graph classification and regression datasets consist of node- and edge-attributed graphs. In my understanding the proposed refinement algorithm can only handle graphs that contain discrete node and edge attributes. This is a serious limitation of the proposed algorithm since it practically means that the algorithm cannot be applied to many interesting learning problems. I wonder whether the algorithm could be somehow generalized to also handle such continuous attributes.

- It is quite surprising that the proposed kernels are compared against only 4 baseline methods, and none of these methods is a graph neural network (GNN). GNNs have attracted a lot of attention recently and have been intensively applied to graph classification tasks. Therefore, I would expect the authors to also compare the proposed kernels against one or more well-established GNNs, e.g., GIN, DiffPool, etc.

- In the introduction of the paper, the authors refer to GNNs a few times, e.g., in lines 29-32 where it is mentioned that the WL algorithm is similar to the idea of neighborhood aggregation in GNNs. In my understanding, it is not trivial to generalize the proposed refinement procedure to GNNs. Furthermore, GNNs produce continuous node features and thus, can already compute, in a sense, finer node similarity measures. However, one would expect the authors to discuss whether and how this procedure could be generalized to GNNs. Since GNNs are the state-of-the-art approach for machine learning on graphs, this is something that would deserve further investigation and in my view it would strengthen a lot the paper.

- I also wonder how sensitive the proposed algorithm is to the initialization of $k$-means. For instance, for graphs whose nodes are not assigned discrete features, the colors that emerge in the first iteration of the refinement procedure correspond to the different degrees of the nodes. Suppose we have the following set of degrees {1,2,3,4,5,6,7,8,9,10} and $k=2$. Suppose, we run the algorithm twice. In the first case $k$-means returns the following two clusters: {1,2} and {3,4,5,6,7,8,9,10}. In the second case, it outputs the following clusters: {1,2,3,4,5} and {6,7,8,9,10}. How performance could vary from one case to the other, and how can one ensure that the optimal performance will be achieved?

Overall, in my view this is an interesting paper which proposes a novel algorithm. However, due to reasons discussed above, I think the work in its current state is unlikely to have an impact on the graph representation learning community, thus I cannot recommend acceptance.

---

### Official Review · Reviewer_2RKG · 2022-10-19

**Overall Score:** 8
**Confidence:** 5

**Review:**

**Summary**:
This paper proposes to adapt the Weisfeiler Leman color refinement (WL) algorithm by replacing the injective (HASH) update function with a non-injective one and through that achieves a more refined node similarity. The original expressiveness of WL is still preserved, because the final stable coloring is the same. Efficient implementations and a practical instance of the new non-injective update function --- $k$-means --- are discussed.
Finally, they show strong empirical performance on standard graph classification tasks and additionally also on the task of approximating the graph edit distance, where they outperform SOTA.



**Main review**:  *Natural and interesting problem, well written, strong theoretical & empirical results*

This is an easy to follow and well written paper. The authors tackle an important problem (the injective, i.e., "yes/no" differentiation of colors through WL is often too coarse to be used as a reasonable similarity function in learning tasks) and provide a general strong framework as a solution. The method is systematically evaluated and achieves strong empirical results. The additional application to graph-edit-distance is surprising and interesting, as well. Also, the implementation is very efficient, in particular, compared to similar variants of WL relying on the computation of the tree-edit distance. I have no doubt that this paper will open up many further additional interesting challenges and contributions as indicated in the conclusion. I recommend accepting this submission to the LoG conference.

**Questions**:

* The choice of $k$-means as a non-injective mapping seems slightly ad hoc. Are there other interesting possibilities e.g., density-based ones? Also, the additional hyperparameter $k$ might be difficult to select practically.
* Is there a way to unify graph/tree edit similarity based WL adaptations and the non-injective approach here? What if the colors have a natural ordering or a metric on them. E.g., the original degree-based colors provides a natural similarity of colors: node with degrees 5 and 6 are probably more similar than 5 and 100.
* Is it possible to adapt the idea of non-injective update functions to GNNs? Perhaps a differentiable alternative to $k$-means is required then.
* Could the transductive nature of this approach (e.g., applying $k$-means also on the test-set graphs and using this information in the learning process) give some small advantage against other approaches that only learn on the training set itself? Also, does this mean that the learning process has to be started all over once a new graph has to be predicted (i.e., apply WL to this new disjoint union of graphs)? Could there be a fast way to perform such an update?

**Further minor remarks**:

* line 186: $|V(G)|-1$ instead of $|V(G)-1|$
* definition of edges $E\subseteq V\times V$ seems like the edges are directed, while later you say it's undirected. Why are you not using the more standard notation for undirected edges $\\{u,v\\}$ and $E\subseteq \\{e \subseteq V \mid |e| = 2\\}$, often denoted as $V \choose 2$.
* the set of labels $L$ is only defined in line 88 but used multiple times before.
* Colors in figures (e.g., 1) are hard to distinguish in grey-scale.
* line 100-101: "relation defines a partial ordering on the colorings", I assume up to $\equiv$ - equivalence classes ( as a poset must be antisymmetric)?
* lines 102-105 are somewhat difficult to understand, please, reformulate or explain with more details.
* paragraph after line 115: maybe also cite the original WL paper (1968)?
* line 147 and 179: "." after V(G). Typo? Might be a ":".
* Theorem 4: The variable total number of iterations $i$ was previously called height $h$ (line 126). Maybe stick to $h$?
* line 22: while it is well-known that strongly regular graphs cannot be distinguished by WL, it might still make sense to cite this fact.
* Maybe make references to e.g., condition 4 clickable (e.g., by defining a new "theorem" enviroment "condition" for the lines 146-149 to make it clearer where it refers to.

---

### Official Review · Reviewer_WUPu · 2022-10-20

**Overall Score:** 5
**Confidence:** 5

**Review:**

### **Summary**:
In this paper, the authors propose a modified Weisfeiler Leman (WL) algorithm for graph similarity computation, dubbed as Gradual WL (GWL). In particular, the authors claim that the WL algorithm is sensitive to local (neighbourhood) graph perturbations, due to the injectivity of the WL update hash function (which in turn leads to fast colour stabilisation) and therefore (intuitively) it might lead to a coarse similarity measure.

To address this, the authors propose to relax the injectivity condition and instead provide distinct updates to only a subset of candidate node colours at each WL iteration. In practice, the choice of the distinct updates is done by a clustering algorithm in the neighbour colour histogram space. In particular, the algorithm is as follows: (1) consider a set of nodes that are assigned with the same colour at iteration $i$, (2) for each of these nodes represent the multiset of the colours of its neighbours as a colour histogram, (3) cluster the colour histograms, (4) assign distinct colours to the nodes corresponding to each cluster at iteration $i+1$. The method is evaluated empirically on graph classification tasks and on a Graph Edit Distance (GED) approximation task, and in several cases, it outperforms methods based on vanilla WL stabilisation.

### **Strengths**:
- **Relevance & Scalability**: Given the strong influence that WL has had in the Graph ML community, it is important to understand its limitations. With that in mind, the present manuscript poses an interesting question, that of the reliability of WL-induced graph similarity, which might be of further interest to the community (beyond graph kernels).  At the same time, the authors propose an approach which is more scalable compared to other graph kernels that have attempted to address the same issue in the past.
- **Presentation**: Overall the paper is well-written, the notation is consistent throughout and most of the concepts are clearly and rigorously explained.

### **Weaknesses**:

**Motivation, claims & ad-hoc options**:

My main concern is that, although the observation of the WL limitation is sound, in my opinion, the proposed methodology does not address it very convincingly. In particular, the approach is mostly ad-hoc and I had trouble finding a rigorous motivation (a theoretical claim or a clear-cut intuition) regarding what we “gain” by (1) the GWL framework in general, (2) the clustering algorithm instantiation in particular. In addition, the experiments, although extensive, don’t make a strong point in favour of this method for a particular use case, since it is unclear what the evaluation metric should be (graph similarity approximation? generalisation in graph ML tasks? sample efficiency?) and some necessary comparisons are missing (see “Evaluation” below). In particular, I have the following concerns:

 -  **Why non-injective?** First off, in my point of view, it might be the case that the injectivity per se is not problematic, rather than its combination with a discrete, un-ordered output space (i.e. that of colours) which typically constrains us into using 0-1 pairwise distances when computing similarities. Perhaps, projecting into a continuous metric space (even injectively) might alleviate the problem (since this can potentially allow defining distances proportional to the magnitude of the perturbations)?
 - **Why clustering?** Second, concerning the injectivity, one has infinitely many options from which to choose a non-injective update function. Why did the authors choose a/this clustering algorithm for this?
- **Why not learning/GNNs?** Going one step forward, one might wonder why the authors chose to hand-design the update function in the first place. Now that the injectivity constraint is dropped, it naturally comes to my mind that the update function can be freely parametrised and learned from the data, *which is exactly what GNNs do*. In particular, one should be able to show that GWL can be expressed by a sufficiently expressive GNN. I do believe that there might be advantages in hand-crafting the update function, since we don’t have any guarantee that the GNN will actually learn the desired one (given a finite number of samples and a certain optimisation algorithm), but to put forward such a claim, appropriate experiments are needed (e.g., showing worse sample efficiency, or worse graph similarity approximation). Unfortunately, here there is no comparison whatsoever with GNNs.
- **What do we gain, if not expressivity?** As I mentioned above, it is very likely that GNNs can express GWL. Clearly, expressivity is not all we care for in ML tasks, but the authors do not make any clear claim about the benefits of not being injective/potentially being less expressive. The discussion about the GED approximation is in a good direction, but it is only intuitive since there is no formal claim and the experimental evaluation is unclear. Also, to make this claim stronger, one needs to understand why being able to approximate the GED is useful for ML tasks.

**Evaluation**:

- **Ablations**: Since the main claim in this paper is that GWL can produce finer similarity measures, I believe that this should be clearly demonstrated in the experimental section, even for artificial tasks. For example, one can think of an experiment where graphs are artificially perturbed and then different similarity measures are evaluated, discussing if they decline proportionally to the magnitude of the perturbations. Similar ideas can be employed for vertex similarity comparisons.
- **GED approximation**: This experiment might be useful to make the point above (although a bit more involved), but its execution is unclear. Why do the authors use k-NN classification accuracy instead of the actual GED approximation? Why didn’t the authors compare against [21], where the only difference is that the colour hierarchy is given by the vanilla WL algorithm? This would be a fairer comparison and would provide deeper insights regarding the differences between the two similarity measures.
- **Comparisons**: As mentioned above, in my opinion, a necessary experiment is a comparison against other, potentially non-injective (or with non-categorical outputs) update functions, with GNNs being the most tangible example. The results against other graph kernels (I assume that they were chosen on the basis of being more “similarity-friendly”) are promising, but not conclusive.
- **Discussion**: Given that beyond the similarity intuition, the paper lacks a convincing claim regarding why one should use GWL for ML tasks, a more extensive analysis of the experimental results is expected: Is this method more expressive (i.e., does it achieve lower training error)? Is the method more sample efficient (i.e., does it generalise better when the training set is smaller)? Currently, the evidence is insufficient.
- It might be also useful to further ablate the choice of this particular update function (e.g., by comparing it against other clustering algorithms).

### **Questions**:
- **Theory**:
    - *I am not sure I understand the claims in 4.1.* In particular, it’s not clear to me why GWL will reach the same stable colouring as WL. Proposition 3 from Ref [18] asserts that there exists a unique coarsest stable colouring, but the refinement operation doesn’t necessarily have to be a local neighbourhood operation (equivalent to what the authors refer to as “renep”), e.g., as far as I understood, even the trivial colouring where every node has a unique colour is considered a refinement. Could the authors explain this part?
    - How does GWL compare against vanilla WL in terms of its expressivity (more precisely w.r.t. its graph isomorphism-distinguishing capacity)?

### **Minor**:
- *Related Work*: I believe that the authors should expand their related work discussion. Currently, only different methods for computing graph similarities based on the WL colours are discussed.
- L161-165: It is not clear to me if GWL is run simultaneously for all graphs in the training set as well. This might be of importance here since the clustering of the colours will be altered.
- L260: Is it correct to say that “vertex assignment … gaining an upper bound of the GED”? Are there cases where we can get an upper bound instead of an approximation? Wouldn’t this imply that this vertex assignment can be used to solve isomorphism (assuming isomorphic graphs get the same vertex assignment)?
 - I am not sure if a conclusive answer to Q2 has been provided. What are the number of iterations and the number of clusters a user should use? How sensitive is the method to these parameters and what factors affect this?
- Is there any clear advantage in favour of GWL over GWLOA or vice-versa?
- Some experimental details are missing: How did the authors split the data for the nested cross-validation and how many splits were used? Is the classifier an SVM and if yes, what parameters have been tuned?
- For completeness, the authors might want to consider also comparing against other types of kernels that are not based on WL (such as the graphlet kernel, Shervashidze et al., AISTATS'09).
- *Technical*: I also found puzzling that the authors obtained improved accuracy in some cases where the WL kernel was computed with a larger number of iterations than those needed for stabilisation. If I understand correctly, the only difference between the kernels after stabilisation is that the vertex matching at the stable colouring is multiplied by a different value (= #iters after stabilization + 1). Does this mean that in some cases it’s beneficial to put more focus only on the stable colouring? This reminds me of the skip-connections from intermediate layers typically used in GNN instantiations, in which the intermediate values are multiplied by a learnable weight matrix, thus allowing the network to learn how to balance between different colourings/features.
- Figure 7 in the appendix is a bit cluttered.


**Suggestions**:

- Perhaps provide an algorithm (in pseudocode) for GWL as this will help to illustrate the method.
- I think it would be helpful if the authors explained in detail how they approximate the GED with GWL, so that the paper is self-contained (now the reader needs to consult Ref. [21] in order to understand). Similarly, for the WLOA kernel (the formula is not provided).


### **Overall**:

The method proposed by the authors provides a way to construct a finer similarity measure compared to the one induced by the WL algorithm, without big sacrifices in the computational complexity. However, (1) it is mainly based on ad-hoc choices, (2) it lacks sufficient and rigorous motivation as to why it is a “better” similarity measure (beyond simple intuition) and why this is helpful for ML tasks, (3) although it seems that it can be expressed by vanilla GNNs, this is not discussed or compared experimentally, which makes it unclear why one should prefer it instead of a GNN. Therefore, at this stage, I am leaning towards rejection.


### **Post-rebuttal**:

My final recommendation remains unchanged, voting for rejection. Please refer to my final comments below for a summary of my reasoning after the discussion with the authors.

---

### Meta-Review · Area_Chair_NNAm · 2022-11-17

**Confidence:** 4
**Recommendation:** Reject

**Meta Review:**

The authors propose a twist on the WL refinement so that similar graphs receive similar encodings. During the refinement process, colors are clustered to remove the injectivity of the standard re-coloring method. Some theoretical results show that a stable coloring is still reached. This was shown to be able to approximate gold standard similarity measure of Graph Edit Distance under a tree metric and shows competitive performance with other graph kernels on classification tasks.

Reviewers find that this is a novel and well-presented method, and that the non-injective nature of the GWL could be relevant for graph comparison. But there is no consensus for how this method would perform against GNNs in classificaton tasks (another popular non-injective method) as no comparison to GNNs is given. Reviewers are split on the importance of this comparison. While the authors do not claim superiority to GNNs in terms of classification, the remaining avenue of novelty which is the approximation of graph similarities would require further evaluation and misses comparisons to related work. More specifically, the proposed procedure has strong conceptual overlap with prior work which also proposes doing a coarser version of WL (via hashing) in order to avoid the strict exact matching of the WL refinement algorithm (Very closely related: [1,2], related and should be cited: [3,4]). The authors should weaken their claim that the gradual WL refinement is a new approach since these other approaches did this as well, and instead focus on what distinguishes their contribution from this prior work. The novel contribution appears to be based on the specific way one chooses to do this gradual version of WL (instead of using LSH, the authors use a k-means based clustering). The authors should cite the relevant related work, discuss their contributions / differences relative to them, and include at least [1,2] in their experiments for a fair assessment of this proposed method.

For these reasons I believe the paper cannot be accepted in its current form and requires further evaluation and positioning within the larger kernel literature.

[1] Neumann, Marion, et al. “Efficient graph kernels by randomization.” Joint European Conference on Machine Learning and Knowledge Discovery in Databases. Springer, Berlin, Heidelberg, 2012.

[2] Shervashidze, Nino. Scalable graph kernels. Diss. Universität Tübingen, 2012., (Section 3.7.3.2)

[3] Morris, Christopher, et al. “Faster kernels for graphs with continuous attributes via hashing.” 2016 IEEE 16th International Conference on Data Mining (ICDM). IEEE, 2016.

[4] Neumann, Marion, et al. “Propagation kernels: efficient graph kernels from propagated information.” Machine Learning 102.2 (2016): 209-245.

---

### Decision · Program_Chairs · 2022-11-23

**Decision:**

Accept (Poster)

**Comment:**

We discussed this case among the PCs and find that the paper is ready for publication since all the major concerns have been addressed during the rebuttal phase. We strongly encourage authors to extend their paper, in particular with respect to the related work, during the camera-ready phase.